# Trajectories of posttraumatic growth and posttraumatic depreciation: A one-year prospective study among people living with HIV

**Małgorzata Pięta⊙\*, Marcin Rzeszutek**

Faculty of Psychology, University of Warsaw, Warsaw, Poland

\* mj.pieta@uw.edu.pl

## Abstract

### Objective

Posttraumatic growth (PTG) and posttraumatic depreciation (PTD) are considered two sides of growth after trauma. Nevertheless, previous studies pointed out that in trauma living with a life-threatening illness, they may be experienced as two independently and share distinct predictors. In our study we aimed to find the different trajectories of PTG and PTD among a sample of people living with HIV (PLWH) and to investigate its predictors out of psychological resilience, and gain and loss of resources from the conservation of resources theory (COR).

### Methods

We designed a longitudinal study that consisted of three measurements at 6-month intervals, and we recruited, respectively, 87, 85 and 71 PLWH. Each time participants filled out the following questionnaires: the expanded version of the PTG and PTD Inventory (PTGDI-X), the Brief Resilience Scale (BRS), the Conservation of Resources Evaluation (COR-E), and a survey on sociodemographic and medical data.

### Results

We observed two separate trajectories of PTG and PTD within participants and found that each of the trajectories were related to different predictors from the studied variables. More specifically, we found a positive relationship between resilience and a descending PTD trajectory that stabilized over time. Gain of resources generally predicted a PTG trajectory, while loss of resources predicted the dynamics of PTD.

### Conclusions

Including two parallel constructs, i.e., PTG and PTD, confirmed the independence of their mechanisms in growth processes among PLWH. The initial insight concerning the role of resilience and resources in PTG/PTD processes may inspire more effective planning for

**Data Availability Statement:** All relevant psychological data are within the article and its Supporting information files, i.e. the values behind the means, standard deviations and other

measures reported, the values used to build graphs that support its main findings. Data provided with the article is anonymized, restricted to psychological variables as it was collected from a patient group and the full dataset contains sensitive medical and sociodemographic data. Additional data are available from the University of Warsaw ethics committee at "etykabadannaukowych@psych.uw.edu.pl" and from Prof. Ewa Gruszczyńska, SWPS University at "egruszczynska@swps.edu.pl" for researchers who meet the criteria for access to confidential data.

**Funding:** the National Science Centre PRELUDIUM 19 grant no. 2020/37/N/HS6/00046.

**Competing interests:** The authors have declared that no competing interests exist.

psychological help for PLWH, and it may stimulate studies on growth after trauma to further examine the two sides of this phenomenon.

## Introduction

The posttraumatic growth (PTG) construct was introduced to the academic world about a 25 years ago, allowing for the empirical examination of a phenomenon deeply ingrained in human nature: growth through trauma and adversity [1, 2]. PTG is most commonly described as an increased sense of self-reliance or strength, increased quality of relationships that includes more compassion and feeling of connectedness, finding a new or different path in life, a greater appreciation for life and spiritual and existential changes. Despite the vast philosophical background for that concept and the numerous theoretical models that emerged over the years, the PTG research area still needs to overcome several important challenges [3, 4]. Recent systematic reviews and meta-analyses [5–8] call for advancements in PTG study designs, including the need for more widespread longitudinal measures of PTG. To this date, very few longitudinal PTG studies have been conducted, and most of these ignore the construct of posttraumatic depreciation (PTD)—defined as reduced of psychological adjustment, impaired cognitive functioning, and low emotional awareness—enabling a parallel measurement of negative changes co-occurring with domains of PTG [9, 10]. Consequently, their results are exposed to potential positivity bias (i.e., overestimation of growth phenomena among participants) [11]. To our best knowledge, only one study has followed the prospective framework in examining PTG and PTD, leaving a significant research gap that calls to be filled [12]. In addition, recent data shows that to gain comprehensive insight into growth dynamics, one should utilize the person-centered approach which assumes the existence of various PTG trajectories among people who are exposed to the same traumatic event but have different psychosocial characteristics [13, 14]. However, until now no research on PTG/PTD using both longitudinal design and the person-centered approach has been conducted. In our study we combined these two methodological designs by studying trajectories of PTG and PTD among people living with HIV [PLWH; 14, 15].

PTG among individuals struggling with life-threatening illnesses, such as cancer, cardiovascular diseases or HIV/AIDS, has been studied almost since this field of research was established [15–17]. Extensive body of research offered important insights into growth processes among these populations, linking PTG phenomena to better well-being or health-related benefits among various patient groups. Nevertheless, PTG studies among patients coping with life-threatening illness remains a challenging area due to the ambiguous nature of illness-related trauma [17]. This latter problem is especially visible among PLWH, whose medical condition can induce psychological distress linked to different areas of their lives and occurring at different stages of HIV infection [18]. The literature on PTG among PLWH is full of inconsistent results regarding its association with sociodemographic data or HIV-related medical variables [19, 20], particularly the time since HIV diagnosis [18, 21, 22]. Also, most studies identify higher PTG levels in women living with HIV, at least in the Western context, and the relationships between PTG and ethnicity, age, and sexual orientation vary depending on the context of a given study [8]. Moreover, HIV-related psychological distress can be characterized by complex etiology and dynamics, particularly when enhanced by social factors, such as HIV/AIDS stigma [8, 23]. Nonetheless, what differentiates somatic threat from a conventional traumatic stressor in its classical sense (e.g., war, natural disaster) [24, 25] is predominantly its internal and chronic nature. Consequently, the PTG triggering factor for PLWH is not universal and

can converge with HIV infection diagnosis or occur many years after [15]. Thus, it is very difficult to discern which events or individual characteristics can promote or prevent PTG within this patient group. What may help us get closer to an answer on the abovementioned research question are advancements in study methodology, such as longitudinal design following the person-centered approach. This approach enables us to extract subgroups of participants characterized by the greatest internal homogeneity and intergroup diversity [26]. It is therefore possible to detect trajectories of different magnitudes of PTG within such heterogenic populations as PLWH [15]. Such design is now widely used for studying patterns of adaptation depending on protective and risk factors examined within a given study group [13]. Nevertheless, very few prospective studies on PTG among PLWH have tried to capture the unique trajectories of this phenomena in this population [14]. And until now, no research on the mutual coexistence of both positive and negative changes in PTG and PTD in a prospective framework have been investigated in these patients.

Consequently, in our study we examined unique predictors of independent PTG/PTD trajectories from psychological variables such as resilience and the levels of resources, as well as sociodemographic and HIV-related clinical variables. In particular, we focused on the ambiguous relationship of resilience with PTG among PLWH as to this date studies indicate to positive, negative and no relationship between these two variables [22, 27]. We followed the operationalization of resilience as an individual ability enabling a person to bounce back or recover from stress and trauma [28]. In addition to resilience, which is an intraindividual characteristic of a person, we also considered a more objective measure of adaptation, i.e., gain vs. loss of psycho-social resources due to living with HIV. Namely, we referred to the conservation of resources (COR) theory, which underlines the role of resources in adverse, as well as traumatic life events [29]. Including COR resources fits well within the ongoing scientific debate on the primarily objective or subjective nature of PTG and its significance for well-being after experiencing trauma [30, 31].

Regarding the abovementioned research gaps in PTG, as well as HIV literature, the aim of our study was twofold. Firstly, we wanted to investigate patterns of PTG vs. PTD change in a 1-year prospective study among PLWH. More specifically, we wanted to verify the potential existence of independent trajectories of these two variables and thus to check if they may constitute independent theoretical constructs [9, 12, 32]. Secondly, we aimed to examine psychological (resilience, level of resources' gain and loss), sociodemographic and clinical correlates of belongingness to these trajectories. To our best knowledge there are no prospective studies following the person-centered approach and examining both PTG and PTD among PLWH. Thus, our study is explorative to a large extent. However, based on one longitudinal study on PTG and PTD [12] and a few prospective studies applying the person-centered approach to PTG [13, 33], including also PLWH [14] we formulated the following research hypotheses:

*Hypothesis 1*. There is a heterogeneity of change in PTG vs. PTD levels (i.e., different classes of trajectories of PTG and PTD can be observed during the study period).

*Hypothesis 2*. PTG/PTD trajectories share different predictors from within the studied variables (see hypotheses 3 and 4).

*Hypothesis 3*. Resilience and resource gains in the first measurement are positively related to an upward PTG trajectory versus a descending PTD trajectory, respectively. Resource loss in the first measurement is inversely linked (i.e., with a descending PTG trajectory and upward PTD trajectory).

*Hypothesis 4*. The extracted PTG/PTD trajectories differ also with respect to measured sociodemographic and HIV-related clinical variables.

## Method

### Participants and procedure

Our study was conducted among PLWH recruited from patients at the Hospital for Infectious Diseases in Warsaw, Poland. The group was invited to participate in a longitudinal study that consisted of three measurements conducted at 6-month intervals. For the first set of measurements that took place in the first half of 2021, we recruited 87 participants. During the initial measurement, they completed an informed consent form by signing on paper a standardized document provided by the University administration and agreed to share personal data and provide their email address or telephone number as a means of communication during the subsequent parts of the study. Also at the first measurement, participants were invited to fill out a paper-and-pencil version of the psychometric questionnaires, including the sociomedical survey. For the second and the third measurements that were conducted consequently in the second half of 2021 and at the beginning of 2022, we prepared electronic versions of the questionnaire using the Google Forms platform. Participants were invited to complete the questionnaire with a message that included a survey link sent via email or SMS. We also enabled patients to continue the study participating in the study using the paper-and-pencil version of the survey that we sent on request via traditional mail containing a return envelope. The tools used were the same for all three measurements, excluding the tool for resilience measurement which we operationalized as stable personality trait. For the subsequent parts of the study, we recruited 85 participants in the second measurement and 71 in the third; thus, 81.6% participated in all three measurements. The drop-out rate was 18.4% and can be explained among others by no remuneration for the study participation and decreased interest and motivation for participating in a repeated measurement for the third time. Table 1 presents the demographic characteristics of the sample.

According to the test based on likelihood ratio, there was no relationship between drop-out and participants' gender, $\lambda(1) = .01$, $p > .05$, being in a stable relationship, $\lambda(1) = .01$, $p > .05$, education, $\lambda(3) = 4.52$, $p > .05$, employment status, $\lambda(3) = 3.28$, $p > .05$, financial status, $\lambda(4) = 5.29$, $p > .05$, sexual orientation, $\lambda(2) = 2.54$, $p > .05$, addiction, $\lambda(1) = 3.03$, $p > .05$, AIDS diagnosis, $\lambda(1) = 1.56$, $p > .05$ or detectable viral load, $\lambda(4) = .09$, $p > .05$. Student's t-tests for independent samples revealed no statistical differences between those participating in all three measurements and drop-out participants regarding age, $t(85) = -.01$, $p > .05$ and years of antiretroviral treatment (ARV), $t(85) = -.34$, $p > .05$.

All the data—contact information, as well as completed questionnaires—were stored on external data disks provided by the University of Warsaw. Participation in the study was voluntary, with no remuneration provided. The eligibility criteria for the study included being at least 18 years of age, having a medical HIV infection diagnosis and entering ARV treatment. Participants were also assessed by medical doctors working in the hospital where the study was held for cognitive disorders constituting exclusion criteria for the study participation. Our study was approved by the local ethics committee.

### Measures

**Expanded version of the PTG and PTD Inventory (PTGDI-X).**   PTG/PTD levels were measured with the 50-item PTGDI-X [34] questionnaire in a validated Polish adaptation. PTGDI-X consists of items evaluating domains in a positive direction of PTG (five subscales: relating to others, new possibilities, personal strength, spiritual change and appreciation of life, for example, I am more willing to express my emotions) accompanied by the same items formulated in a negative way to assess PTD (e.g., I am less willing to express my emotions).

**Table 1. Demographic characteristics of the study sample.**

| | | First measurement | | Two measurements | | All three measurements | |
|---|---|---|---|---|---|---|---|
| | | n | % | n | % | n | % |
| Gender | Women | 16 | 18.4 | 16 | 18.8 | 13 | 18.3 |
| | Men | 71 | 81.6 | 69 | 81.2 | 58 | 81.7 |
| Age | | 20–73 | M = 41.13; SD = 11.09 | 20–73 | M = 40.82; SD = 10.88 | 20–73 | M = 41.13; SD = 11.52 |
| Relationship | In stable relationship | 42 | 48.3 | 41 | 48.2 | 36 | 50.7 |
| Education | Primary | 1 | 1.1 | 1 | 1.2 | 1 | 1.4 |
| | Vocational | 4 | 4.6 | 3 | 3.5 | 4 | 5.6 |
| | Secondary | 32 | 36.8 | 32 | 37.6 | 23 | 32.4 |
| | Higher | 50 | 57.5 | 49 | 57.6 | 43 | 60.6 |
| Employment | Regular employment | 58 | 66.7 | 57 | 67.1 | 45 | 63.4 |
| | Unemployed | 12 | 13.8 | 12 | 14.1 | 11 | 15.5 |
| | Pension | 13 | 14.9 | 13 | 15.3 | 11 | 15.5 |
| | Retired | 4 | 4.6 | 3 | 3.5 | 4 | 5.6 |
| Financial status | Very good | 13 | 14.9 | 12 | 14.1 | 12 | 16.9 |
| | Good | 38 | 43.7 | 37 | 43.5 | 29 | 40.8 |
| | Medium | 28 | 32.2 | 28 | 32.9 | 22 | 31.0 |
| | Bad | 5 | 5.7 | 5 | 5.9 | 5 | 7.0 |
| | Very bad | 3 | 3.4 | 3 | 3.5 | 3 | 4.2 |
| Sexual orientation | Heterosexual | 20 | 23.0 | 20 | 23.5 | 14 | 19.7 |
| | Homosexual | 58 | 66.7 | 56 | 65.9 | 50 | 70.4 |
| | Other | 9 | 10.3 | 9 | 10.6 | 7 | 9.9 |
| Addiction | Addicted | 18 | 20.7 | 18 | 21.2 | 12 | 16.9 |
| AIDS | Diagnosis | 17 | 19.5 | 15 | 17.6 | 12 | 16.9 |
| Viral load | Detectable | 7 | 8.0 | 7 | 8.2 | 6 | 8.5 |
| ARV treatment | In years | .6–30 | M = 7.22; SD = 5.34 | .6–30 | M = 7.32; SD = 5.35 | .6–30 | M = 7.31; SD = 5.54 |

Participants respond on a 6-point scale ranging from 0 (I did not experience this change) to 5 (I experienced this change to a great degree). Higher scores are a sign of more intense PTG or PTD levels. We followed the global PTG and PTD scores according to the recommendation of Taku et al. [34]. Participants were instructed to concentrate on the positive or negative changes in their lives after receiving their HIV diagnosis. The Cronbach's alphas for the global PTG and PTD scores can be found in Table 2.

**The Brief Resilience Scale (BRS).** Resilience, defined as the ability to "bounce back" in the aftermath of stressful life events, was evaluated with the Polish adaptation of the BRS [28] scale by Konaszewski [39]. BRS is a short, 6-item scale with a 5-point Likert response scale (1 —strongly disagree—to 5—strongly agree). The Cronbach's alphas for this tool can be found in Table 2.

**Conservation of Resources Evaluation (COR-E).** Resource gain and loss were evaluated with the aid of the short version of the COR-E questionnaire [29] in the validated Polish adaptation. COR-E consists of 40 items describing resources related to family, power, vitality, wealth, and spirituality. Participants are describing the extent to which they experienced gains or losses in these resources on a Likert scale (0—no change—to 5—a very large loss/gain). Two main indicators were constructed, one for resource gain and the other for loss. Participants were asked to report their subjective gain or loss of resources following the moment of diagnosis of their HIV infection. The Cronbach's alphas for the COR-E can be found in Table 2.

**Table 2. Descriptive statistics for analyzed variables.**

| Variables | M | SD | min | max | S | K | α |
|---|---|---|---|---|---|---|---|
| Resilience | 20.41 | 5.45 | 6 | 30 | -.31 | -.39 | .85 |
| Gain | 1.36 | 1.23 | 0 | 5 | .67 | -.56 | .97 |
| Loss | 0.61 | 0.89 | 0 | 5 | .10 | .53 | .97 |
| PTG | | | | | | | |
| Meas. I | 52.32 | 34.03 | 0 | 125 | .04 | -.16 | .97 |
| Meas. II | 49.02 | 32.20 | 0 | 115 | .17 | -.11 | .97 |
| Meas. III | 47.17 | 31.62 | 0 | 112 | .16 | -.22 | .97 |
| PTD | | | | | | | |
| Meas. I | 22.40 | 25.25 | 0 | 115 | .29 | .08 | .95 |
| Meas. II | 25.54 | 24.91 | 0 | 94 | .99 | .07 | .96 |
| Meas. III | 27.24 | 25.51 | 0 | 104 | .01 | .47 | .96 |

*Note*: Meas—Measurement; M—mean value; SD—standard deviation;

min—minimum value; max—maximum value; S—skewness;

K—kurtosis; α—Cronbach's α reliability coefficient.

## Data analysis

In the preliminary analysis, descriptive statistics and Pearson correlation coefficients were calculated. Next, latent class growth analysis (LCGA) [35] was used to extract subgroups of respondents with different trajectories of changes in PTG and PTD. The use of this method enables identifying homogeneous subpopulations within the larger heterogeneous population [35]. In our study we assessed four models: a model with one general trajectory for the whole sample, a model with two different trajectories, a model with three different trajectories and a model with four different trajectories. The model with the lowest value of Bayesian Information Criterion (BIC) fit index was chosen on the condition that extracted profiles were detected in at least 20% of cases in the sample. The levels of resilience and gain and loss of resources in the first measurement were analyzed as predictors of detected types of trajectories with the use of logistic regression analysis. The extracted classes representing different trajectories were then compared in terms of participants' age, gender, employment, addiction, AIDS diagnosis and sexual orientation. Statistical significance was verified with the Student's t-test for independent samples and a statistical test based on likelihood ratio.

## Results

Table 2 presents descriptive statistics for analyzed variables. It shows mean values, standard deviations, minimum and maximum values and the values of skewness and kurtosis. The values of skewness and kurtosis did not exceed the range from -1.0 to 1.0. Therefore, parametric statistical tests were used in the subsequent analysis.

Table 3 presents the values of Pearson's correlation coefficients between analyzed variables. Statistically significant correlations are marked with asterisks.

The main analysis was based on latent class growth analysis. Table 4 presents the values of BIC index values and profiles distribution for all analyzed models. The models with best fit (i.e., the lowest value of BIC fit index and extracted profiles detected in at least 20% of the sample) are marked with a bold font.

**Table 3. Correlation coefficients between analyzed variables.**

| | Variables | 1. | 2. | 3. | 4. | 5. | 6. | 7. | 8. |
|---|---|---|---|---|---|---|---|---|---|
| 1. | Resilience | - | - | - | - | - | - | - | - |
| 2. | Gain | .096* | - | - | - | - | - | - | - |
| 3. | Loss | -.323** | .057 | - | - | - | - | - | - |
| | PTG | | | | | | | | |
| 4. | Meas. I | -.009 | .633** | .137** | - | - | - | - | - |
| 5. | Meas. II | .098 | .382** | .018 | .584** | - | - | - | - |
| 6. | Meas. III | .035 | .297* | .022 | .583** | .814** | - | - | - |
| | PTD | | | | | | | | |
| 7. | Meas. I | -.385** | .050 | .538** | .151** | .077 | .161 | - | - |
| 8. | Meas. II | -.245* | .051 | .575** | .133 | .134 | .245* | .655** | - |
| 9. | Meas. III | -.277* | -.018 | .529** | .078 | .081 | .294* | .560** | .865** |

*Note*: Meas.—Measurement;

* p < .05;

** p < .01.

## PTG and PTD trajectories

Two different trajectories were extracted in the analyses of PTG and PTD. Fig 1 depicts extracted trajectories for PTG. In analysis of PTG, a trajectory of growth (profile 1) and a trajectory of decrease (profile 2) were detected.

Fig 2 depicts extracted trajectories for PTD. In analysis of PTD, a trajectory of growth (profile 1) and a trajectory of decrease (profile 2) were also detected.

## Predictors of PTG and PTD trajectories

The relationships between the levels of resilience, gain and loss in the first measurement and detected trajectories were analyzed with the use of logistic regression analysis. Resilience and resource gain and loss were analyzed as predictors. The types of PTG and PTD trajectory (profile 1 or profile 2) were analyzed as explained variables. The results are presented in Table 5. The decreasing trajectories (profile 2) were coded as 1.

**Table 4. BIC index values and profiles distribution for analyzed models.**

| Variables | No. of profiles | BIC | Frequency distribution for trajectories | | | |
|---|---|---|---|---|---|---|
| | | | Profile 1% | Profile 2% | Profile 3% | Profile 4% |
| PTG | 1 | 6555.67 | 100 | | | |
| | **2** | **6419.53** | **51.7** | **48.3** | | |
| | 3 | 6424.82 | 1.6 | 50.9 | 47.5 | |
| | 4 | 6419.51 | 16.8 | 40.0 | 41.6 | 1.6 |
| PTD | 1 | 6177.92 | 100 | | | |
| | **2** | **5968.40** | **19.7** | **80.3** | | |
| | 3 | 5901.93 | 72.6 | 7.3 | 20.1 | |
| | 4 | 5881.98 | 15.6 | 3.7 | 12.2 | 68.4 |

*Note*: BIC—Bayesian Information Criterion.

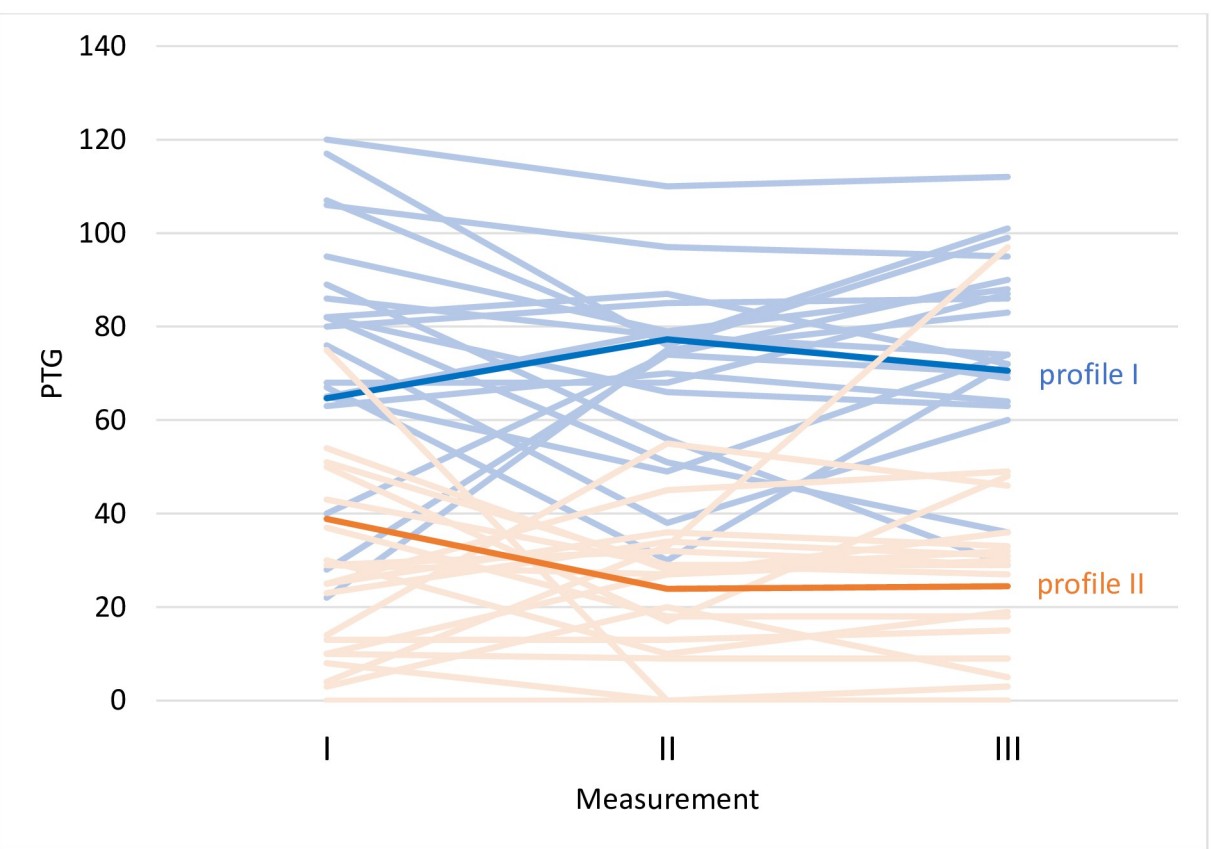

**Fig 1. Detected profiles of trajectories regarding changes in PTG.** The two extracted profiles detected in at least 20% of cases in the sample are highlighted (i.e., the model with the lowest value of Bayesian Information Criterion (BIC)).

**Resilience.** There were statistically significant relationships between levels of resilience in the first measurement and the trajectory of PTD. The higher the level of resilience in the first measurement, the higher the odds of a decreasing trajectory of PTD (profile 2). The acquired results are consistent with hypothesis 2; there was, however, no statistically significant relationship between the level of resilience in the first measurement and the type of PTG trajectory.

**Resources.** The level of resource gain in the first measurement was significantly related to the trajectory of PTG. The level of loss in the first measurement was significantly related to the trajectory of PTD. The higher the level of gain in the first measurement, the higher the odds of an increasing trajectory of PTG (profile 1). The higher the level of loss in the first measurement, the higher the odds of an increasing trajectory of PTD (profile 1). The acquired results are consistent with hypothesis 2; there was, however, no statistically significant relationship between the level of gain in the first measurement and the type of PTD trajectory and no statistically significant relationship between the level of loss in the first measurement and the type of PTG trajectory.

**Sociodemographic and medical variables.** The extracted subgroups of respondents with different trajectories were compared in terms of participants' age, gender, employment, addiction, AIDS diagnosis, and sexual orientation. The mean age of participants with an increasing trajectory of PTG (profile 1) was 40.24 ($SD = 10.66$). The mean age of participants with a decreasing trajectory of PTG (profile 2) was 39.95 ($SD = 10.19$). According to the t-test value of independent samples, the difference was not statistically significant, $t(505) = .31$, $p > .05$.

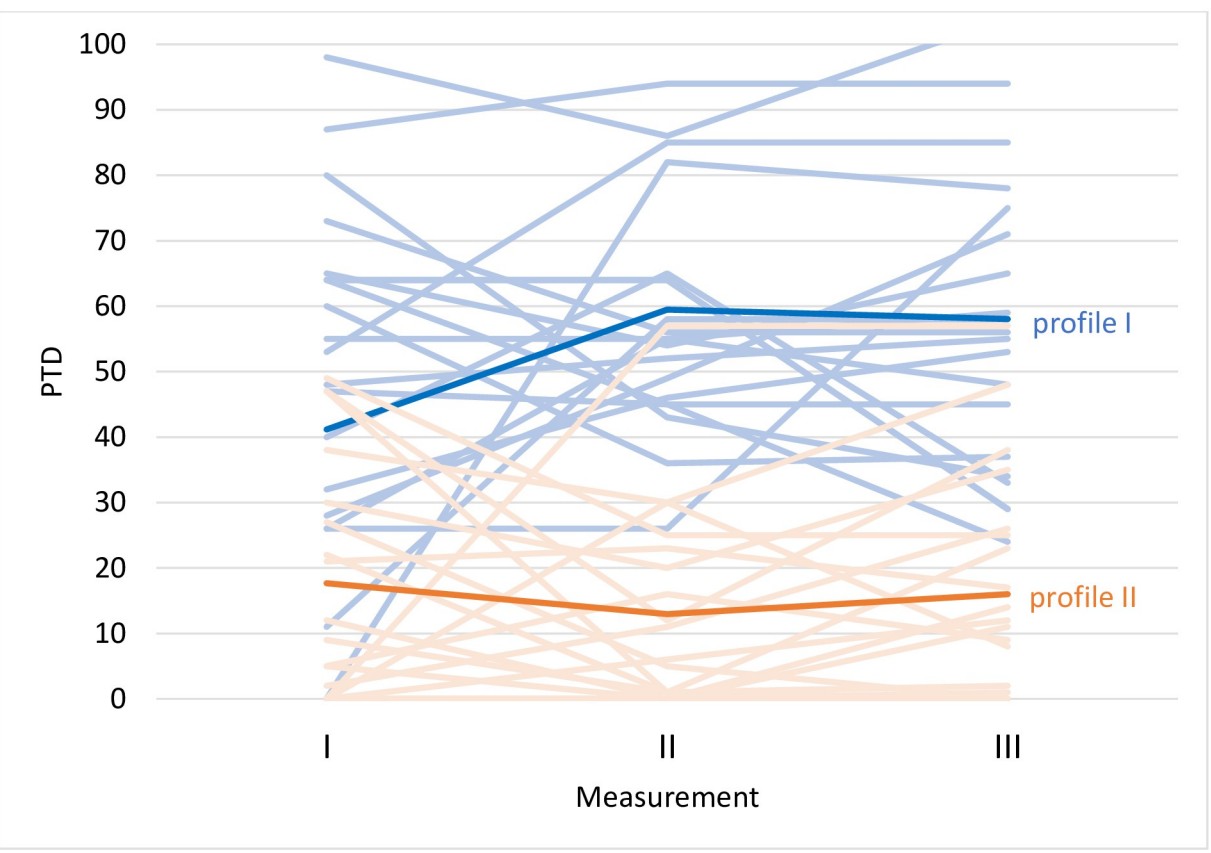

**Fig 2. Detected profiles of trajectories regarding changes in PTD.** The two extracted profiles detected in at least 20% of cases in the sample are highlighted (i.e., the model with the lowest value of Bayesian Information Criterion (BIC)).

The mean age of participants with an increasing trajectory of PTD (profile 1) was 42.35 ($SD$ = 11.63). The mean age of participants with a decreasing trajectory of PTD (profile 2) was 39.54 ($SD$ = 10.04). According to the t-test value of independent samples, the difference was statistically significant, $t(137.49)$ = 2.22, $p < .05$. Participants with an increasing trajectory of PTD were significantly older than participants with a decreasing trajectory of PTD. Table 6 presents the distribution of participants' gender, employment, addiction, AIDS diagnosis and sexual orientation in the subgroups of participants with detected trajectories of change of PTG and PTD with values of statistical test based on likelihood ratio.

**Table 5. Analysis of relationships between the levels of resilience, gain and loss in the first measurement and trajectories of PTG and PTD among study participants.**

| Explained trajectory | Predictor | *OR* | *Wald* | *df* | *p* |
|---|---|---|---|---|---|
| PTG | Resilience | 0.99 | 0.39 | 1 | .535 |
| | Gain | 0.64 | 29.98 | 1 | .001 |
| | Loss | 1.00 | 0.00 | 1 | .987 |
| PTD | Resilience | 1.08 | 12.80 | 1 | .001 |
| | Gain | 1.12 | 1.38 | 1 | .240 |
| | Loss | 0.54 | 25.73 | 1 | .001 |

*Note*: *OR*—odds ratio; *Wald*—Wald test for significance of predictor;

*df*—degrees of freedom; *p*—statistical significance.

**Table 6. Distributions of demographic characteristics in subgroups of participants with different trajectories regarding PTG and PTD.**

| | | Profile 1 | | Profile 2 | | λ | df | p |
|---|---|---|---|---|---|---|---|---|
| | | n | % | n | % | | | |
| Trajectories based on PTG | | | | | | | | |
| Gender | Women | 32 | 12.2% | 35 | 14.3% | .47 | 1. | .491 |
| | Men | 230 | 87.8% | 210 | 85.7% | | | |
| Employment | Regular employment | 189 | 72.1% | 185 | 75.5% | 1.40 | 3 | .706 |
| | Unemployed | 29 | 11.1% | 28 | 11.4% | | | |
| | Pension | 32 | 12.2% | 23 | 9.4% | | | |
| | Retired | 12 | 4.6% | 9 | 3.7% | | | |
| Addiction | Addicted | 39 | 14.9% | 40 | 16.3% | .20 | 1 | .655 |
| AIDS | Diagnosis | 49 | 18.7% | 33 | 13.5% | 2.57 | 1 | .109 |
| Sexual orientation | Heterosexual | 70 | 26.7% | 65 | 26.5% | 8.04 | 2 | .018 |
| | Homosexual | 160 | 61.1% | 167 | 68.2% | | | |
| | Other | 32 | 12.2% | 13 | 5.3% | | | |
| Trajectories based on PTD | | | | | | | | |
| Gender | Women | 14 | 14.0% | 53 | 13.0% | .07 | 1 | .797 |
| | Men | 86 | 86.0% | 354 | 87.0% | | | |
| Employment | Regular employment | 59 | 59.0% | 315 | 77.4% | 13.73 | 3 | .002 |
| | Unemployed | 17 | 17.0% | 40 | 9.8% | | | |
| | Pension | 16 | 16.0% | 39 | 9.6% | | | |
| | Retired | 8 | 8.0% | 13 | 3.2% | | | |
| Addiction | Addicted | 22 | 22.0% | 57 | 14.0% | 3.90 | 1 | .048 |
| AIDS | Diagnosis | 24 | 24.0% | 58 | 14.3% | 5.19 | 1 | .023 |
| Sexual orientation | Heterosexual | 25 | 25.0% | 110 | 27.0% | 10.98 | 1 | .004 |
| | Homosexual | 57 | 57.0% | 270 | 66.3% | | | |
| | Other | 18 | 18.0% | 27 | 6.6% | | | |

The number of participants with regular employment was significantly lower in the group of participants with an increasing trajectory of PTD (profile 1). The number of addicted participants was significantly higher in the group of participants with an increasing trajectory of PTD (profile 1). The number of participants with a diagnosis of AIDS was also significantly higher in the group of participants with an increasing trajectory of PTD (profile 1). The number of participants with homosexual orientation was significantly lower in the group of participants with an increasing trajectory of PTG (profile 1). The number of participants with homosexual orientation was also significantly lower in the group of participants with an increasing trajectory of PTD (profile 1).

## Discussion

Since PTG and its predictors remain a matter of ongoing scientific discussion that supports the search for methodological improvements in this research area [5–8], the primary goal of our study was to test the newest extension of this term in the form of concepts of PTG and PTD supported by the longitudinal study design in the clinical sample of PLWH.

### Independent trajectories of growth and depreciation in PLWH

The results we obtained confirmed the first two hypotheses concerning independence of PTG and PTD constructs and stating differences within their predictors [12, 32]. Specifically, we

detected two independent trajectories within PTG and in PTD (Figs 1 & 2), and within both we found that resilience and resource gains and losses predicted their paths that we will discuss further on. Also, by means of LCGA [35], we found parallel dynamics within the PTG and PTD trajectories (Figs 1 & 2). During the first period of the study (i.e., between the first and the second measurement), we detected ascending and descending changes of direction within PTG and PTD, which stabilized during the following study period. These results remain in line with previous research findings and provide evidence for independence and coexistence of PTG and PTD processes within clinical samples [32]. At the same time, the added value provides the first insights from longitudinal, person-centered data concerning the abovementioned relationships between the two constructs and the potential mechanisms underlying positive and negative change. Most importantly, our results speak for multidimensional rather than two-dimensional consequences of coping with health-related trauma by proving that PTG and PTD can be observed either simultaneously or separately, or neither of these may be reported [12, 32].

## Predictors of growth and depreciation in PLWH

**Resilience.** The results of the study confirmed our third hypothesis concerning PTG/PTD trajectory predictors in the form of resilience up to a point. Although we did not detect a significant relationship between initial resilience levels and PTG change dynamics in neither ascending nor descending trajectories, the role of resilience in predicting PTD was confirmed in accordance with our predictions. Specifically, the resilience level was associated positively with a descending PTD trajectory, whereas an inverse trend was observed for the trajectory characterized by an ascending dynamic (Fig 2). This role of resilience in predicting PTD levels is probably the most interesting part of our study results and can be potentially treated as an important complementary argument in ongoing discussion concerning the role of resilience in posttraumatic growth process. To date, the significance of resilience in PTG is considered to be ambiguous: firstly, at the level of various possible operationalizations of this construct in the general population after trauma and adversity (see resilience as a process or resilience as a personality trait) [36, 37] and secondly, concerning the direction of its association with PTG, especially among PLWH [22, 27, 38]. Some studies on PLWH highlight the role of resilience as a potential shield against HIV-related trauma and as an actual improvement in various areas of life, implying learning and growing from this kind of adversity [22]. However, as HIV becomes less of a medical burden, in some cases returning to baseline functioning may be more relevant than growth form HIV diagnosis, the relationship between resilience and the opposite of growth, i.e., PTD, may be of greater importance [8]. In our study we addressed this issue by joining longitudinal study design with a person-centered approach and framing resilience as an innate ability to "bounce back" from adverse life circumstances [28, 39] that may be of particular importance for PLWH [27, 40]. The results constitute a valuable addition to the ongoing discussion concerning the role of resilience in PTG process by speaking for its role as a protective factor against the PTD dynamic, rather than a trait supporting PTG.

**Resource loss and gain.** We also examined the role of resource gains and losses, operationalized according to COR theory, as possible predictors of PTG/PTD trajectories. Although exploratory, this hypothesis was confirmed but only to a certain degree. We found gains and losses associated irrespectively to either PTG or PTD trends (Figs 1 & 2) (i.e., initial resource gain promoting an ascending PTG dynamic), along with an analogous trend for initial resource loss within a PTD trajectory (Fig 2). This simultaneous occurrence of fluctuations within changes in resources and changes in PTG/PTD dynamics speaks for considering PTG/PTD and resource gain/loss as overlapping constructs, especially as our results were obtained

within a study that measured consequences of long-term trauma exposure. Specifically, the parallel use of COR-E and PTGDI-X measures in the context of a life-threatening somatic condition confirms the view of PTG as an unequivocally salutogenic posttrauma outcome [30, 31]. This is in line with the standpoint of Hobfoll et al. [30], who argued for the need to measure two sides of posttraumatic change (i.e., to discern between objective or subjective posttraumatic outcomes). Still, more research is needed to fully understand differences between subjective/illusory PTG and actual positive change as well as their significance for well-being [31, 41]. Nevertheless, the current research can inspire further inquiry into the significance of actual changes for entering growth or depreciation dynamics, especially in the framework of longitudinal, person-centered perspectives.

**Sociodemographic and clinical variables.** In accordance with our last hypothesis, we found the obtained PTG/PTD trajectories differed to a large degree in respect of sociodemographic and HIV-related clinical characteristics. Firstly, we saw that people who were entering an ascending PTD trajectory were older than people who showed a descending trend in this trajectory (Fig 2). Although this result refers to the PTD dynamic, it is in line with other studies that show a negative association between age and PTG in samples of participants belonging to sexual minorities as elevated stigma and worse well-being are observed predominantly in this group of PLWH [8]. This result may be also specific for Polish population of PLWH as the results concerning the role socio-demographic characteristics of PLWH in PTG vary greatly across different contexts [8]. Moreover, our finding can be associated with greater health-related anxiety in this group as preoccupation with one's somatic condition can increase with age and medical advancement of HIV-treatment is still a very recent phenomenon. Also, PTG is a phenomenon that assumes openness to new opportunities, and both expectancy and zeal for such experiences may decrease with age and be replaced with resignation, which can be reflected in the PTD dynamic. Further, we observed fewer fully employed participants within the increasing PTD trajectory (Fig 2). This is also in accordance with previous study results in PLWH samples, where employment, as well as education, showed relatively homogenous positive effects on well-being [8, 18]. This trend can reflect a lower level of stigma and its isolating effects among participants who continue to actively participate in society despite their diagnosis. Also, we observed a higher number of homosexual participants within trajectories characterized by both descending PTG and PTD trend (Figs 1 & 2). As our study was held in a capital city and within the major HIV clinic in the country, this result may reflect a unique effect of the community culture that was previously observed in numerous nonprofit organizations serving PLWH, which frequently target gay or bisexual men living with HIV [8]. Lastly, we also observed an intuitive positive association between active substance misuse, as well as entering the AIDS phase of HIV infection, and following the ascending PTD trajectory (Fig 2). This last result is also in line with previous HIV research [8, 18]; however, because they were collected from self-reports, they should be considered with caution. Nevertheless, it can be hypothesized that nonadaptive coping that impedes ARV treatment adherence, as well as other comorbidities, can support entering an ascending PTD trajectory.

## Strengths and limitations

The longitudinal research design with three consecutive measurements of both PTG and PTD, combined with a person-centered approach to data analysis, was a major strength of our study. It was also the first study to apply such advanced methodology with the purpose of adding to an understanding of the ambiguous relationship between resilience, resources and PTG processes among a clinical sample suffering from chronic stress or trauma linked to HIV diagnosis. Nevertheless, our study was not free of limitations. First and foremost, the participants

differed significantly regarding time since receiving an HIV diagnosis, which can additionally complicate understanding of PTG-triggering process within a population that is already unorthodox regarding this area of study. Future studies should focus on recruiting more homogenous samples of PLWH to characterize the influence of different trauma-inducing events on PTG trajectories. Secondly, the final group of study participants was relatively small and female participants were outnumbered by male participants. Such a demographic structure of the study sample could lead to a low ability to detect heterogeneity of trajectories, as it doesn't adequately reflect the gender ratio of people diagnosed with HIV/AIDS (UNAIDS, 2022). Finally, our participants were a relatively highly functional population characterized by immune health parameters, with mostly undetectable viral loads. This population characteristic can be an additional source of bias for our study [42], although previous research suggests that associations between medical characteristics of PLWH are less important for the self-reported PTG than psychosocial characteristics [18]. Nevertheless, future studies should include more reliable measures of medical characteristics to uncover possible relationships between immune and psychological functioning of PLWH.

## Conclusions

Overall, our study highlighted many important aspects of PTG and PTD processes among PLWH. In particular, the unique role of resilience as a protective factor against entering a trajectory of posttraumatic depreciation in this population may be accepted as its major finding. This result was obtained by implementing an advanced study methodology and by using two separate scales for measuring both positive and negative aspects of change in the aftermath of a trauma. This study design also confirmed previous study results, such as the independence of PTG and PTD among clinical samples, but also showed different predictors for PTG and PTD phenomena and their respective dynamics among extracted subgroups of participants. Although further research is needed to fully explain the unique dynamics of PTG and PTD among PLWH, it seems that avoiding depreciation rather than searching for growth may be the most adaptive strategy for maintaining psychological well-being among this population. This observation is of major importance for providing effective psychological help adjusted to PLWH' individual characteristics and needs [18]. It seems that, as an HIV diagnosis has in most cases no significant influence on the objective health status of people living with the virus, a favorable environment may make it possible to treat HIV infection as a minor rather than traumatic stressor. Also, the prevailing negative master narrative associated with HIV in the Polish context requires PLWH to constantly cope with HIV/AIDS stigma and prevents them from building on the resilient capacity to experience personal growth and translate their experience into a universal and relatable source of growth.

## Supporting information

**S1 Dataset.**
(XLSX)

## Author Contributions

**Conceptualization:** Małgorzata Pięta, Marcin Rzeszutek.

**Data curation:** Małgorzata Pięta.

**Formal analysis:** Marcin Rzeszutek.

**Funding acquisition:** Małgorzata Pięta.

**Investigation:** Małgorzata Pięta.

**Methodology:** Małgorzata Pięta, Marcin Rzeszutek.

**Project administration:** Małgorzata Pięta, Marcin Rzeszutek.

**Resources:** Małgorzata Pięta.

**Software:** Marcin Rzeszutek.

**Supervision:** Marcin Rzeszutek.

**Validation:** Marcin Rzeszutek.

**Visualization:** Małgorzata Pięta, Marcin Rzeszutek.

**Writing – original draft:** Małgorzata Pięta.

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
