## [Decision Letter · Decision Letter 0]

22 Jul 2022

PONE-D-22-09045Trajectories of posttraumatic growth and posttraumatic depreciation:A one-year prospective study among people living with HIVPLOS ONE

Dear Dr. Pięta,

Thank you for submitting your manuscript to PLOS ONE. After careful consideration, we feel that it has merit but does not fully meet PLOS ONE’s publication criteria as it currently stands. Therefore, we invite you to submit a revised version of the manuscript that addresses the points raised during the review process.

Please note that we have only been able to secure a single reviewer to assess your manuscript. We are issuing a decision on your manuscript at this point to prevent further delays in the evaluation of your manuscript. Please be aware that the editor who handles your revised manuscript might find it necessary to invite additional reviewers to assess this work once the revised manuscript is submitted. However, we will aim to proceed on the basis of this single review if possible.

The reviewer raised a number of concerns that need attention. They request additional information on methodological aspects of the study (such as the inclusion of information on the reasons of the drop-out rate), and request for the result and discussions section to be re-organized, with results better integrated in the discussion section. 

Could you please revise the manuscript to carefully address all concerns raised?

We look forward to receiving your revised manuscript.

Kind regards,

Katrien Janin, Phd

Staff Editor

PLOS ONE

Journal Requirements:

“This project was funded from the National Science Centre PRELUDIUM 19 grant no. 2020/37/N/HS6/00046.”

“the National Science Centre PRELUDIUM 19 grant no. 2020/37/N/HS6/00046.”

Reviewers' comments:

Reviewer's Responses to Questions

**Comments to the Author**

1. Is the manuscript technically sound, and do the data support the conclusions?

Reviewer #1: Yes

2. Has the statistical analysis been performed appropriately and rigorously? 

Reviewer #1: Yes

3. Have the authors made all data underlying the findings in their manuscript fully available?

Reviewer #1: No

4. Is the manuscript presented in an intelligible fashion and written in standard English?

Reviewer #1: Yes

5. Review Comments to the Author

Reviewer #1: General comment

The present study aimed to identify the different trajectories of posttraumatic growth (PTG) and posttraumatic depreciation (PTD) constructs and their possible predictors in a group of people living with HIV (PLWH). In order to reach this goal, a longitudinal study was carried out with three evaluation times (baseline, 6 months later, and 1 year later).

The paper deals with a clinically interesting topic, given the various trajectories that PTG and PTD may assume over time, based on different individual characteristics. However, I have some specific comments for improvement of the article that are reported below.

Specific comments:

1) In the introductive section, I would suggest the authors to include a definition of both PTG and PTD dimensions, in order to make those constructs clearer to the reader.

Moreover, I would suggest the authors to dedicate more space to the presentation of the available data on PTG/PTD in people leaving with HIV, reducing perhaps the first part of this section. Indeed, the association between PTG and individual characteristics is immediately exposed, without introducing this topic in the specific context of HIV.

Finally, in this section have been presented some methodological aspects that could be only mentioned here and deepened in the next sections of the manuscript. Similarly, the importance of assessing specific psychological factors in the context of HIV (i.e., resilience and resources) could be better highlighted.

2) With regard to the study aim, I would suggest the authors to integrate this subparagraph with the main body of the introduction, in order to make this section easier to be followed.

3) In the participants’ description, I would suggest the authors to describe the main reasons of drop-out rate. Moreover, the time period in which the evaluation has been carried out could be included.

4) With regard to the measures administered in the present study, it is not clear if the authors employed an ad hoc Polish translation of each questionnaire or if validated and published Polish versions of those instruments are available.

5) I am not an expert of latent class growth analysis; however, the analyses seem to be adequately performed. I would only suggest the authors to include some sub-paragraphs (in the result section) in which the main results of the study are reported in a more organized way.

6) Similar to the result section, I would suggest that the authors better organize the discussion of the main findings of the study (e.g., the following sentence “... and within both we found that resilience and resource gains and losses predicted their paths that we will discuss further on” could be confusing for the reader).

In addition, findings could be better integrated and discussed in relation to the previous evidence that investigated the present psychological constructs both in people with HIV and other medical conditions.

7) In the conclusive section of the manuscript, I would suggest the authors to include some clinical implications of the present findings.

8) Please correct some typing errors throughout the manuscript.

6. PLOS authors have the option to publish the peer review history of their article (what does this mean?). If published, this will include your full peer review and any attached files.

Reviewer #1: No

---

## [Author Response · Author response to Decision Letter 0]

16 Aug 2022

Dear Reviewer, 

thank you very much for suggestions and remarks concerning our article titled “Trajectories of posttraumatic growth and posttraumatic depreciation: A one-year prospective study among people living with HIV”, which we would like to publish in PLOS ONE. We have introduced all remarks mentioned in the review. Below we cite the specific remark and, in bold and in parentheses, our answer to the Reviewer’s remarks. All the changes in the manuscript were tracked and marked in yellow.

1. Is the manuscript technically sound, and do the data support the conclusions?

Reviewer #1: Yes

[Thank you very much for approval of our reasoning in this manuscript.]

2. Has the statistical analysis been performed appropriately and rigorously? 

Reviewer #1: Yes

[Thank you very much for analysing and approving the statistical methods we chose.]

3. Have the authors made all data underlying the findings in their manuscript fully available?

Reviewer #1: No

[Thank you for this remark. We provided an anonymized dataset with our revision. All additional data regarding our datasets will be made available upon request.]

4. Is the manuscript presented in an intelligible fashion and written in standard English?

Reviewer #1: Yes

[Thank you for approving our language. The manuscript was professionally proofread.]

5. Review Comments to the Author

Reviewer #1: General comment

The present study aimed to identify the different trajectories of posttraumatic growth (PTG) and posttraumatic depreciation (PTD) constructs and their possible predictors in a group of people living with HIV (PLWH). In order to reach this goal, a longitudinal study was carried out with three evaluation times (baseline, 6 months later, and 1 year later).

The paper deals with a clinically interesting topic, given the various trajectories that PTG and PTD may assume over time, based on different individual characteristics. However, I have some specific comments for improvement of the article that are reported below.

Specific comments:

1) In the introductive section, I would suggest the authors to include a definition of both PTG and PTD dimensions, in order to make those constructs clearer to the reader.

[Thank you for this remark. We provided more exhaustive definitions of the two phenomena.]

Moreover, I would suggest the authors to dedicate more space to the presentation of the available data on PTG/PTD in people leaving with HIV, reducing perhaps the first part of this section. Indeed, the association between PTG and individual characteristics is immediately exposed, without introducing this topic in the specific context of HIV.

[Thank you for this point. We elaborated on PTG phenomenon across studies concerning PLWH.]

Finally, in this section have been presented some methodological aspects that could be only mentioned here and deepened in the next sections of the manuscript. Similarly, the importance of assessing specific psychological factors in the context of HIV (i.e., resilience and resources) could be better highlighted.

[Thank you for this remark. We elaborated more on the importance of assessing resilience and resources in the context of HIV.]

2) With regard to the study aim, I would suggest the authors to integrate this subparagraph with the main body of the introduction, in order to make this section easier to be followed.

[Thank you for this remark. We merged the two sections.]

3) In the participants’ description, I would suggest the authors to describe the main reasons of drop-out rate. Moreover, the time period in which the evaluation has been carried out could be included.

[Thank you for this point. We added information on possible drop out reasons and study period.]

4) With regard to the measures administered in the present study, it is not clear if the authors employed an ad hoc Polish translation of each questionnaire or if validated and published Polish versions of those instruments are available.

[Thank you for this point. We clarified the tool description.]

5) I am not an expert of latent class growth analysis; however, the analyses seem to be adequately performed. I would only suggest the authors to include some sub-paragraphs (in the result section) in which the main results of the study are reported in a more organized way.

[Thank you very much for this comment. We not only restructured the result section, but also included a clearer description of LCGA in the data analysis section.]

6) Similar to the result section, I would suggest that the authors better organize the discussion of the main findings of the study (e.g., the following sentence “... and within both we found that resilience and resource gains and losses predicted their paths that we will discuss further on” could be confusing for the reader).

[Thank you for this point. We appreciate your comment on this section. According to your suggestion, to make our discussion clearer to the reader we restructured the section.]

In addition, findings could be better integrated and discussed in relation to the previous evidence that investigated the present psychological constructs both in people with HIV and other medical conditions.

[Thank you for this remarked. We clarified and broadened our discussion in the context of previous findings.]

7) In the conclusive section of the manuscript, I would suggest the authors to include some clinical implications of the present findings.

[Thank you very much for this remark. We highlighted clinical, as well as wider social implication of our study.]

8) Please correct some typing errors throughout the manuscript.

[Thank you very much for this point. We did our best to find and correct the errors.]

6. PLOS authors have the option to publish the peer review history of their article (what does this mean?). If published, this will include your full peer review and any attached files.

Do you want your identity to be public for this peer review? For information about this choice, including consent withdrawal, please see our Privacy Policy.

Reviewer #1: No

---

## [Decision Letter · Decision Letter 1]

9 Sep 2022

Trajectories of posttraumatic growth and posttraumatic depreciation:

A one-year prospective study among people living with HIV

PONE-D-22-09045R1

Dear Dr. Pięta,

We’re pleased to inform you that your manuscript has been judged scientifically suitable for publication and will be formally accepted for publication once it meets all outstanding technical requirements.

Kind regards,

Sónia Brito-Costa, Ph.D.

Academic Editor

PLOS ONE

Additional Editor Comments (optional):

Reviewers' comments:

Reviewer's Responses to Questions

**Comments to the Author**

1. If the authors have adequately addressed your comments raised in a previous round of review and you feel that this manuscript is now acceptable for publication, you may indicate that here to bypass the “Comments to the Author” section, enter your conflict of interest statement in the “Confidential to Editor” section, and submit your "Accept" recommendation.

Reviewer #1: (No Response)

2. Is the manuscript technically sound, and do the data support the conclusions?

Reviewer #1: (No Response)

3. Has the statistical analysis been performed appropriately and rigorously? 

Reviewer #1: (No Response)

4. Have the authors made all data underlying the findings in their manuscript fully available?

Reviewer #1: (No Response)

5. Is the manuscript presented in an intelligible fashion and written in standard English?

Reviewer #1: (No Response)

6. Review Comments to the Author

Reviewer #1: (No Response)

7. PLOS authors have the option to publish the peer review history of their article (what does this mean?). If published, this will include your full peer review and any attached files.

Reviewer #1: No

---

## [Editor Report · Acceptance letter]

13 Sep 2022

PONE-D-22-09045R1 

Trajectories of posttraumatic growth and posttraumatic depreciation: A one-year prospective study among people living with HIV 

Dear Dr. Pięta:

I'm pleased to inform you that your manuscript has been deemed suitable for publication in PLOS ONE. Congratulations! Your manuscript is now with our production department. 

Kind regards, 

on behalf of

Dr. Sónia Brito-Costa 

Academic Editor

PLOS ONE